# On Spectral Vectorial Differential Equation of Generalized Hermite Polynomials

**Mohamed Jalel Atia [1,*] and Majed Benabdallah [2]**

1   Department of Mathematics, College of Sciences, Qassim University, Buraydah 51452, Saudi Arabia
2   Laboratory of Mathematics and Applications (LR17ES11), Faculty of Sciences of Gabes, University of Gabès, Gabès 6072, Tunisia; benabdallahmajed@gmail.com
*   Correspondence: jalel.atia@gmail.com

**Abstract:** In this paper, we first give some results on monic generalized Hermite polynomials (GHP) $\{H_n^{(\mu)}(x)\}_{n \geq 0}$, orthogonal with respect to the positive weight $|x|^{2\mu}e^{-x^2}$, $\mu > -\frac{1}{2}$, $x \in \mathbb{R}$, which will lead to the formulation of the second-order spectralvectorial differential equation (SVDE) that the GHP satisfies. This SVDE differs from the one given in G. Szego (problem 25. p. 380), which is a pseudo-spectral equation. Second, we give the SVDE, as conjecture, satisfied by the generalized Jacobi polynomials $J_n^{(\alpha,\alpha+1)}(x,\mu)$, orthogonal with respect to the positive weight $w(x,\alpha;\mu) = |x|^{-\mu}(1-x^2)^{\alpha}(1-x)$, $\mu < 1$, $\alpha > -1$ on $[-1,1]$.

**Keywords:** classical orthogonal polynomials; semiclassical orthogonal polynomials; second order linear differential equation

**MSC:** 42C05; 33C45





## 1. Introduction

In 1929, Böchner [1] classified all classical orthogonal polynomial (COP) $\{P_n\}_{n \geq 0}$ solutions of a second-order Sturm–Liouville differential equation of the form

$$\phi(x)D_x^2 P_n(x) - \psi(x)D_x P_n(x) = \lambda_n P_n(x), \ n \geq 0, \tag{1}$$

where $D_x = \frac{d}{dx}$, $\phi$ and $\psi$ are polynomials such that $deg \ \phi \leq 2$ and $deg \ \psi = 1$ and $\lambda_n \neq 0$ is the eigenvalue associated with the eigenvector $P_n(x)$.

If we denote the linear operator $\phi(x)D_x^2 - \psi(x)D_x$ by $\Xi$ then for any COP $P_n$ associated with the eigenvalue $\lambda_n$ we have

$$\Xi^{(.)}P_n(x) = \lambda_n^{(.)}P_n(x),$$

as follows

- Hermite polynomial $H_n$, $n \geq 0$:

$$\Xi^{(h)} := D_x^2 - 2xD_x; \lambda_n^{(h)} = -2n. \tag{2}$$

- Laguerre polynomial $L_n^{(\alpha)}$, $n \geq 0$:

$$\Xi^{(l)} := xD_x^2 - (x - \alpha - 1)D_x; \lambda_n^{(l)} = -n \tag{3}$$

- Bessel polynomial $B_n^{(\alpha)}$, $n \geq 0$:

$$\Xi^{(b)} := x^2 D_x^2 + 2(\alpha x + 1)D_x; \lambda_n^{(b)} = n(n + 2\alpha - 1). \tag{4}$$

- Jacobi polynomial $P_n^{(\alpha,\beta)}, n \geq 0$:

$$\Xi^{(j)} := (x^2 - 1)D_x^2 + ((\alpha + \beta + 2)x + \beta - \alpha)D_x; \lambda_n^{(j)} = n(n + \alpha + \beta + 1). \quad (5)$$

To summarize, all COP fulfil

$$\Xi^{(\cdot)}P_n(x) = \lambda_n^{(\cdot)}P_n(x), \quad (6)$$

$\Xi^{(\cdot)}$ *is a linear operator not depending on n and* $\lambda_n^{(\cdot)}$ *does not depend on x,* (7)

this is to say that COP fulfil;s the property of spectral theory, which makes COP the perfect and main eigenvectors used in many areas of physics, chemistry and other disciplines. This property is no longer true for the generalization of COP, which is called semi-classical orthogonal polynomials (SCOP) because their differential equation is written as [2]

$$A(x,n)P_n''(x) + B(x,n)P_n'(x) + C(x,n)P_n(x) = 0,$$

where $A(x,n)$, $B(x,n)$ and $C(x,n)$ depend on $x$ and $n$.

In 2000, J. Koekoek and R. Koekoek [3] studied the spectral type differential equations satisfied by the generalized Jacobi polynomials, which are orthogonal in the interval $[-1, 1]$ with respect to a weight function consisting of the classical Jacobi weight function together with two point masses at the endpoints of the interval of orthogonality.

In 2010, F. Marcellan et al. [4] dealt with the problem of the second-order pseudo-spectral linear differential equation fulfilled by the symmetric SCOP of class 1 which satisfies:

$$\phi_1(x)D_x^2 P_n(x) + \phi_2(x)D_x P_n(x) = \chi(x,n)P_n(x), \quad (8)$$

where $\phi_i$, $i = 1, 2$ are polynomials with $\phi_2$ monic and the degrees of the polynomials $\chi(.\,,n)$ are uniformly bounded. The authors found $\chi(x,n) = a_n x^2 + b_n$ as follows,

- the Generalized Hermite case: $\chi(x,n) = -2(nx^2 + \mu\eta_n)$,
- the Generalized Bessel case: $\chi(x,n) = n(n+2\nu)x^2 - \frac{1}{2}\eta_n$,
- the Generalized Gegenbauer case: $\chi(x,n) = n(2\alpha + 2\beta + 2 + n)x^2 + (2\beta + 1)\eta_n$, where $\eta_n = \frac{(-1)^n - 1}{2}$, $n \geq 0$.

The expression $\chi(x,n)$ depends on $x$ and, then, (6) and (7) do not hold.

In the present paper, we consider the monic GHP $H_n^{(\mu)}$, $n \geq 0$ introduced by Szego [5], p. 380, Problem 25, as a set of real polynomials orthogonal with respect to the weight $|x|^{2\mu}e^{-x^2}$, $\mu > -\frac{1}{2}$, $x \in \mathbb{R}$. These polynomials were then investigated by Chihara in his Ph.D. thesis [6] and further studied by Rosenblum in [7]. These monic polynomials $H_n^{(\mu)}$ of degree equal to $n$ are defined by [8–17]

$$H_{2n+\varepsilon}^{(\mu)}(x) = (-1)^n \Gamma\left(n + \mu + \frac{1}{2} + \varepsilon\right) \sum_{k=0}^{n} \frac{(-1)^k \binom{n}{k} x^{2k+\varepsilon}}{\Gamma(k + \mu + \frac{1}{2} + \varepsilon)}, \quad (9)$$

where $n \geq 0$, $\varepsilon = 0$ or 1 and $\mu > -\frac{1}{2}$.

Besides (2), the classical Hermite polynomials (CHP) $H_n$, $n \geq 0$, are given by

$$H_n(x) = \frac{n!}{2^n} \sum_{k=0}^{\lfloor \frac{n}{2} \rfloor} \frac{(-1)^k (2x)^{n-2k}}{k!(n-2k)!}, \quad n \geq 0, \quad (10)$$

where $\lfloor x \rfloor$ is the integer part of the real $x$. These polynomials are eigenvectors of the endomorphism $\Xi^{(h)}$ on $\mathbb{R}_n[x]$, presented by the following diagonal matrix $M_H^{(n)}$

$$
M_H^{(n)} = \begin{pmatrix}
0 & 0 & 0 & 0 & \cdots & 0 \\
0 & -2 & 0 & 0 & \cdots & 0 \\
0 & 0 & -4 & 0 & \cdots & 0 \\
0 & 0 & 0 & -6 & \cdots & 0 \\
\vdots & \vdots & \ddots & \ddots & \ddots & 0 \\
0 & 0 & 0 & 0 & \cdots & -2n
\end{pmatrix} = Diag(-2k,\ 0 \le k \le n). \tag{11}
$$

The GHP $H_n^{(\mu)}$, $n \ge 0$, as far as we know, are not eigenvectors of any known $R_n[x]$-endomorphism.

It is a well known fact that the only monic orthogonal polynomial sequence $\{B_n\}_{n \ge 0}$ satisfying the relation $D_x B_{n+1}(x) = (n+1)B_n(x)$, $n \ge 0$, for the ordinary derivative operator $D_x$, is the Hermite sequence, up to an affine transformation [18]. This last relationship defines the so-called Appell sequences [19], which are widely spread in the literature in several contexts and applications. They present a large variety of features and include other famous polynomial sequences, such as the Bernoulli one (which is not an orthogonal polynomial sequence).

The aim of this manuscript is to solve the following problem: give explicitly all the coefficients $\lambda_{k,n}$, $0 \le k \le n$, $n \ge 1$ which fulfill the second-order linear differential equation

$$
\Xi^{(h)}(H_n^{(\mu)}(x)) = \lambda_n^{(h)} H_n^{(\mu)}(x) + \sum_{k=0}^{n-1} \lambda_{k,n} H_k^{(\mu)}(x), \tag{12}
$$

where $\Xi^{(h)} = D_x^2 - 2x D_x$ and $\lambda_n^{(h)} = -2n$. Equation (12) will be called the second-order spectral vectorial differential equation that the GHP satisfies.

The contents of the paper are as follows. In Section 2, we give some preliminary results. In Section 3, we give a new property, which will be needed for the sequel. In Section 4, we give, explicitly, the upper triangular matrix $M_\mu^{(n)}$ presenting the endomorphism $\Xi^{(h)}$ for these generalized classical cases, such that when $\mu = 0$, we recover the diagonal matrix for classical Hermite polynomials. Then, we present the main result as a characterization of the second-order spectral vectorial differential equation that the GHP satisfies. Finally, we give a conclusion for this work and we present, as a conjecture, the second-order spectral vectorial differential equation that a non-symmetric case of generalized Jacobi polynomials satisfies.

## 2. Preliminary Results

First, let $\mathcal{P}$ denote the vector space of polynomials with coefficients in $\mathbb{C}$ and let $\mathcal{P}'$ be it's dual. We denote by $\langle u, p \rangle$ the action of linear functional $u \in \mathcal{P}'$ on $p \in \mathcal{P}$. In particular, $(u)_n = \langle u, x^n \rangle$, $n \ge 0$ are called the moments of $u$.

Let $\{P_n\}_{n \ge 0}$ be a semi-classical sequence orthogonal with respect to $u$ that is to say that $u$ satisfies a functional equation, i.e.,

$$
D(\phi u) + \psi u = 0, \tag{13}
$$

with

$$
\langle D(\phi u), p \rangle = -\langle \phi u, p' \rangle \tag{14}
$$

$$
\langle \psi u, p \rangle = \langle u, \psi p \rangle. \tag{15}
$$

$\phi$ and $\psi$ are the given polynomials related to the sequence $\{P_n\}_{n \ge 0}$.

Recall the three-term recurrence relation (TTRR) satisfied by monic orthogonal polynomials,

$$
P_0(x) = 1, \qquad P_1(x) = x - \beta_0,
$$
$$
P_{n+2}(x) = (x - \beta_{n+1})P_{n+1}(x) - \gamma_{n+1}P_n(x), \tag{16}
$$

with $\beta_n \in \mathbb{C}$, $\gamma_n \in \mathbb{C}^*$ and $\gamma_0 := \langle u, 1 \rangle = 1$.

Among the various characterizations of semi-classical sequences we list the following relations that are equivalent.

(1) Second-order linear differential equation of Maroni type [2].

$$\phi(x)D_x^2 P_n(x) - \psi(x)D_x P_n(x) = M(x,n)P_n(x) - \gamma_n D_x(D_n(x))P_{n-1}(x), \qquad (17)$$

with

$$M(x,n) = \frac{D_x C_n(x) - D_x C_0(x)}{2} + \sum_{\nu=0}^{n-1} D_\nu(x)$$

(2) Second-order differential equation of Laguerre–Perron type [2].

$$\phi(x)D_n(x)D_x^2 P_n(x) + \{C_0(x)D_n(x) - W(\phi(x), D_n(x))\}D_x P_n(x)$$

$$+ \{W\big(\frac{C_n(x) - C_0(x)}{2}, D_n(x)\big) - D_n(x)\sum_{k=0}^{n-1} \frac{D_k(x)}{\gamma_k}\}P_n(x) = 0, \qquad (18)$$

where

$$\begin{cases} C_{n+1}(x) = -C_n(x) + 2\frac{D_n(x)}{\gamma_n}(x - \beta_n), & n \geq 0; \\ D_{n+1}(x) = -\phi(x) + \frac{\gamma_n}{\gamma_{n-1}}D_{n-1}(x) + \frac{D_n(x)}{\gamma_n}(x - \beta_n)^2 - C_n(x)(x - \beta_n), & n \geq 0; \\ C_0(x) = -\phi(x) - \psi(x), \\ D_0(x) = -D_x\big( < u_y, \frac{\phi(x) - \phi(y)}{x - y} > \big) - < u_y, \frac{\psi(x) - \psi(y)}{x - y} >, \\ W(f,g)(x) = f(x)D_x(g(x)) - D_x(f(x))g(x), \end{cases} \qquad (19)$$

Each $P_n$ satisfies the structure relationship given in the following lemma,

**Lemma 1** ([2]). *Let* $\{P_n\}_{n \geq 0}$ *be a monic orthogonal sequence with respect to* $u$, $(u)_0 = 1$ *and let* $(\phi, \psi)$ *a couple polynomials with* $deg\phi = t$, $deg\psi = p$ *and* $s = max(p - 1, t - 2)$. *Therefore, for each* $n \geq 0$ *there is a single polynomial* $q_s(x,n)$ *of degree* $s$ *at the most and a unique system of complex numbers* $\lambda_{n,\nu}$, $0 \leq \nu \leq n$, $n \geq 0$ *such as:*

$$\phi(x)D_x^2 P_{n+1} + \psi(x)D_x P_{n+1} = q_s(x,n)P_{n+1}(x) + \sum_{\nu=0}^{n} \lambda_{n,\nu}P_\nu(x), \ n \geq 0. \qquad (20)$$

## 3. Properties of the Monic GHP

The GHP $H_n^\mu$, $n \in \mathbb{N}$, are defined as a set of real polynomials orthogonal with respect to the weight $|x|^{2\mu}e^{-x^2}$, $\mu > -\frac{1}{2}$, i.e.,

$$< H_n^\mu, H_m^\mu > := \int_{-\infty}^{+\infty} H_n^\mu(x)H_m^\mu(x)e^{-x^2}|x|^{2\mu}dx; n, m \geq 0, \qquad (21)$$

$$< H_{2n+\varepsilon}^\mu, H_{2m+\varepsilon}^\mu > = n!\Gamma(\mu + n + \frac{1}{2} + \varepsilon)\delta_{n,m}, \qquad (22)$$

where $\varepsilon = 0$ or $1$ and $\delta_{n,m}$ is the Kronecker symbol.

These polynomials fulfill the following TTRR:

$$H_0^{(\mu)}(x) = 1, \qquad H_1^{(\mu)}(x) = x,$$

$$H_{n+2}^{(\mu)}(x) = xH_{n+1}^{(\mu)}(x) - \gamma_{n+1}H_n^{(\mu)}(x), \qquad (23)$$

$$where \ \gamma_{2n+\varepsilon} = \mu(2 - \varepsilon) + \frac{2n + \varepsilon}{2}, \ n \geq 0 \ and \ \varepsilon = 1 \ or \ 2. \qquad (24)$$

**Proposition 1.** *For $0 \leq k \leq n-1$, $n \geq 1$ and $\mu > -\frac{1}{2}$, the GHP fulfill*

$$< H_n^{(\mu)}(x), x^k > := \int_{-\infty}^{\infty} |x|^{2\mu} e^{-x^2} H_n^{(\mu)}(x) x^k dx = 0, \tag{25}$$

$$< H_n^{(\mu)}(x), x^n > \neq 0. \tag{26}$$

In this section, we give some properties of GHP, which we will need in the sequel. The first property is

**Proposition 2.**

$$H_{2n+1}^{(\mu)}(x) = x H_{2n}^{(\mu+1)}(x), \ n \geq 0, \tag{27}$$

$$H_{2n+2}^{(\mu)}(x) = H_{2n+2}^{(\mu+1)}(x) + (n+1) H_{2n}^{(\mu+1)}(x), \ n \geq 0. \tag{28}$$

**Proof.** Using (9) for $\varepsilon = 0$ and $\varepsilon = 1$ we obtain

$$H_{2n+1}^{(\mu)}(x) = (-1)^n \Gamma(n + \mu + \frac{1}{2} + 1) \sum_{k=0}^{n} \frac{(-1)^k \binom{n}{k} x^{2k+1}}{\Gamma(k + \mu + \frac{1}{2} + 1)},$$

and

$$H_{2n}^{(\mu)}(x) = (-1)^n \Gamma(n + \mu + \frac{1}{2}) \sum_{k=0}^{n} \frac{(-1)^k \binom{n}{k} x^{2k}}{\Gamma(k + \mu + \frac{1}{2})}.$$

This last equation can be written as

$$H_{2n}^{(\mu+1)}(x) = (-1)^n \Gamma(n + \mu + \frac{1}{2} + 1) \sum_{k=0}^{n} \frac{(-1)^k \binom{n}{k} x^{2k}}{\Gamma(k + \mu + \frac{1}{2} + 1)}.$$

This gives (27).
For the second, we use (23) together with (27), we obtain:

$$H_{2n+3}^{(\mu)}(x) = x H_{2n+2}^{(\mu)}(x) - (n+1) H_{2n+1}^{(\mu)}(x); \quad H_{2n+3}^{(\mu)}(x) = x H_{2n+2}^{(\mu+1)}(x), \ n \geq 0.$$

These two equations give $x H_{2n+2}^{(\mu)}(x) - (n+1) H_{2n+1}^{(\mu)}(x) = x H_{2n+2}^{(\mu+1)}(x)$, $n \geq 0$. Using (27) again and dividing by $x$, we get the desired expression. $\square$

Substituting the parameter $\mu$ by $\mu + 1$, we get the second property:

**Proposition 3.** *For $n \geq 0$ and $\mu > -\frac{1}{2}$, we get*

$$H_n^{(\mu+1)}(x) = \sum_{k=0}^{\lfloor \frac{n}{2} \rfloor} \frac{(-1)^k \lfloor \frac{n}{2} \rfloor!}{(\lfloor \frac{n}{2} \rfloor - k)!} H_{n-2k}^{(\mu)}(x), \ n \geq 0, \tag{29}$$

**Proof.** Using (28) we obtain $H_{2n}^{(\mu+1)} = H_{2n}^{(\mu)} - n H_{2n-2}^{(\mu+1)}$, and recursively, we get

$$H_{2n}^{(\mu+1)} = H_{2n}^{(\mu)} - n(H_{2n-2}^{(\mu)} - (n-1) H_{2n-4}^{(\mu+1)}),$$

then we find

$$H_{2n}^{(\mu+1)} = H_{2n}^{(\mu)} - n H_{2n-2}^{(\mu)} + n(n-1) H_{2n-4}^{(\mu)} - n(n-1)(n-2) H_{2n-6}^{(\mu)}$$
$$+ \ldots + (-1)^k n(n-1) \ldots (n-k+1) H_{2n-2k}^{(\mu)} + \ldots + (-1)^n n! H_0^{(\mu)}.$$

For the odd case, first, use the TTRR given in (23) and we obtain

$$H_{2n+1}^{(\mu+1)}(x) = xH_{2n}^{(\mu+1)}(x) - nH_{2n-1}^{(\mu+1)}(x), \ n \geq 0,$$

then we use (27) and get $H_{2n+1}^{(\mu+1)}(x) = H_{2n+1}^{(\mu)}(x) - nH_{2n-1}^{(\mu+1)}(x), \ n \geq 0$, then, recursively, we get

$$H_{2n+1}^{(\mu+1)}(x) = H_{2n+1}^{(\mu)}(x) - n(H_{2n-1}^{(\mu)}(x) - (n-1)H_{2n-3}^{(\mu+1)}),$$

then we find

$$H_{2n+1}^{(\mu+1)}(x) = H_{2n+1}^{(\mu)}(x) - nH_{2n-1}^{(\mu)}(x) + n(n-1)H_{2n-3}^{(\mu)} - n(n-1)(n-2)H_{2n-5}^{(\mu)}$$
$$+ \ldots + (-1)^k n(n-1) \ldots (n-k+1)H_{2n+1-2k}^{(\mu)} + \ldots + (-1)^n n! H_1^{(\mu)}.$$

□

Now, we give two relationships between the GHP $H_n^{(\mu)}$ and the CHP $H_n$:

**Proposition 4.** *For $n \geq 0$ and $\mu > -\frac{1}{2}$, we have*

$$H_n^{(\mu)}(x) = \sum_{k=0}^{\lfloor \frac{n}{2} \rfloor} (-1)^k \binom{\lfloor \frac{n}{2} \rfloor}{k} \mu^{\overline{k}} H_{n-2k}(x) \tag{30}$$

$$H_n(x) = \sum_{k=0}^{\lfloor \frac{n}{2} \rfloor} \binom{\lfloor \frac{n}{2} \rfloor}{k} \mu^{\underline{k}} H_{n-2k}^{(\mu)}(x) \tag{31}$$

*where $H_n$ is given in (10). The falling factorial $z^{\underline{n}}$ and the Pochhammer symbol $z^{\overline{n}}$ are defined, for $z \in \mathbb{C}$ and $n \in \mathbb{Z}$, as follows*

$$z^{\underline{n}} = x(x-1) \ldots (x-n+1), \tag{32}$$
$$z^{\overline{n}} = x(x+1) \ldots (x+n-1), \tag{33}$$
$$z^{\underline{0}} = z^{\overline{0}} = 1. \tag{34}$$

**Proof.** The proof of (30) will be performd by recurrence using the TTRR (23).

For $n = 0$, (30) gives $1 = 1$, for $n = 1$ (30) give $x = x$ and for $n = 2$ (30) give $x^2 - \frac{\mu+1}{2} = x^2 - \frac{1}{2} - \mu$. Then, for any $n$ we have

$$H_{n+2}^{(\mu)}(x) = xH_{n+1}^{(\mu)}(x) - \gamma_{n+1}H_n^{(\mu)}(x), \ n \geq 0.$$

With the substitution of $n$ by $2n+1$ we get

$$H_{2n+3}^{(\mu)}(x) = xH_{2n+2}^{(\mu)}(x) - (n+1)H_{2n+1}^{(\mu)}(x), \ n \geq 0.$$

Using the recurrence hypothesis, $H_n^{(\mu)}(x) = \sum_{k=0}^{\lfloor \frac{n}{2} \rfloor} (-1)^k \binom{\lfloor \frac{n}{2} \rfloor}{k} \mu^{\overline{k}} H_{n-2k}(x)$, for $2n+2$ and $2n+1$ the rhs becomes

$$x \sum_{k=0}^{n+1} (-1)^k \binom{n+1}{k} \mu^{\overline{k}} H_{2n+2-2k}(x) - (n+1) \sum_{k=0}^{n} (-1)^k \binom{n}{k} \mu^{\overline{k}} H_{2n+1-2k}(x)$$

$$= x(-1)^{n+1} \binom{n+1}{n+1} \mu^{\overline{n+1}} H_0(x) + \sum_{k=0}^{n} (-1)^k \mu^{\overline{k}} (x \binom{n+1}{k} H_{2n+2-2k}(x) - (n+1) \binom{n}{k} H_{2n+1-2k}(x))$$

$$= (-1)^{n+1} \mu^{\overline{n+1}} H_1(x) + \sum_{k=0}^{n} (-1)^k \mu^{\overline{k}} \binom{n+1}{k} (xH_{2n+2-2k}(x) - (n+1-k) H_{2n+1-2k}(x))$$

$$= (-1)^{n+1} \mu^{\overline{n+1}} H_1(x) + \sum_{k=0}^{n} (-1)^k \mu^{\overline{k}} \binom{n+1}{k} H_{2n+3-2k}(x)$$

$$= \sum_{k=0}^{n+1} (-1)^k \mu^{\overline{k}} \binom{n+1}{k} H_{2n+3-2k}(x).$$

With substitution of $n$ by $2n$ in TTRR (23), we get

$$H_{2n+2}^{(\mu)}(x) = x H_{2n+1}^{(\mu)}(x) - (\mu + n + \frac{1}{2}) H_{2n}^{(\mu)}(x), \ n \geq 0.$$

Using

$$H_n^{(\mu)}(x) = \sum_{k=0}^{\lfloor \frac{n}{2} \rfloor} (-1)^k \binom{\lfloor \frac{n}{2} \rfloor}{k} \mu^{\overline{k}} H_{n-2k}(x)$$

the rhs becomes

$$x \sum_{k=0}^{n} (-1)^k \binom{n}{k} \mu^{\overline{k}} H_{2n+1-2k}(x) - (\mu + n + \frac{1}{2}) \sum_{k=0}^{n} (-1)^k \binom{n}{k} \mu^{\overline{k}} H_{2n-2k}(x)$$

$$= \sum_{k=0}^{n} (-1)^k \binom{n}{k} \mu^{\overline{k}} (x H_{2n+1-2k}(x) - (\mu + n + \frac{1}{2}) H_{2n-2k}(x))$$

$$= \sum_{k=0}^{n} (-1)^k \binom{n}{k} \mu^{\overline{k}} (x H_{2n+1-2k}(x) - (n - k + \frac{1}{2}) H_{2n-2k}(x)) - \sum_{k=0}^{n} (-1)^k \binom{n}{k} \mu^{\overline{k}} (\mu + k) H_{2n-2k}(x)$$

$$= \sum_{k=0}^{n} (-1)^k \binom{n}{k} \mu^{\overline{k}} H_{2n+2-2k}(x) - \sum_{k=0}^{n} (-1)^k \binom{n}{k} \mu^{\overline{k+1}} H_{2n-2k}(x)$$

$$= (-1)^0 \binom{n}{0} \mu^{\overline{0}} H_{2n+2}(x) + \sum_{k=1}^{n} (\binom{n}{k} + \binom{n}{k-1}) (-1)^k \mu^{\overline{k}} H_{2n+2-2k}(x) + (-1)^{n+1} \mu^{\overline{n+1}} H_0$$

$$= (-1)^0 \binom{n}{0} \mu^{\overline{0}} H_{2n+2}(x) + \sum_{k=1}^{n} \binom{n+1}{k} (-1)^k \mu^{\overline{k}} H_{2n+2-2k}(x) + (-1)^{n+1} \mu^{\overline{n+1}} H_0$$

$$= \sum_{k=0}^{n+1} (-1)^k \binom{n+1}{k} \mu^{\overline{k}} H_{2n+2-2k}(x).$$

This has the same proof for the second equation. $\square$

Using (30), we can give the transformation matrix $P_\mu^{(n)}$ from the basis $\mathfrak{B} = (H_0, H_1, \ldots, H_n)$ to the basis $\mathfrak{B}' = (H_0^\mu, H_1^\mu, \ldots, H_n^\mu)$, which is an upper triangular matrix. By (31), we deduce $Q_\mu^{(n)}$ the inverse matrix of $P_\mu^{(n)}$.

Let us introduce $P_\mu^{(n)} = (a_{i,j})$, $1 \leq i, j \leq n+1$ with $a_{i,j} = 0$ if $i > j$.
For $i \leq j$:

$$a_{i,j} = \begin{cases} h_{\lfloor \frac{i-1}{2} \rfloor, j-1} & if \ (j-i) \text{ is even,} \\ 0 & if \ (j-i) \text{ is odd,} \end{cases} \tag{35}$$

where

$$h_{k,2p} = h_{k,2p+1} = (-1)^{p-k} \binom{p}{p-k} \mu^{\overline{p-k}}, 0 \leq k \leq p. \tag{36}$$

$Q_\mu^{(n)} = (b_{i,j})$, $1 \leq i, j \leq n+1$ with $b_{i,j} = 0$ if $i > j$.
For $i \leq j$:

$$b_{i,j} = \begin{cases} \tilde{h}_{\lfloor \frac{i-1}{2} \rfloor, j-1} & if \ (j-i) \text{ is even,} \\ 0 & if \ (j-i) \text{ is odd,} \end{cases} \tag{37}$$

where

$$\tilde{h}_{k,2p} = \tilde{h}_{k,2p+1} = \binom{p}{p-k} \mu^{\underline{p-k}}; \ 0 \le k \le p. \tag{38}$$

For example, for $n = 4$ (even) we have:

$$P_\mu^{(n)} = \begin{pmatrix} h_{0,0} & 0 & h_{0,2} & 0 & h_{0,4} \\ 0 & h_{0,1} & 0 & h_{0,3} & 0 \\ 0 & 0 & h_{1,2} & 0 & h_{1,4} \\ 0 & 0 & 0 & h_{1,3} & 0 \\ 0 & 0 & 0 & 0 & h_{2,4} \end{pmatrix} = \begin{pmatrix} 1 & 0 & -\mu & 0 & \mu(\mu+1) \\ 0 & 1 & 0 & -\mu & 0 \\ 0 & 0 & 1 & 0 & -2\mu \\ 0 & 0 & 0 & 1 & 0 \\ 0 & 0 & 0 & 0 & 1 \end{pmatrix}.$$

$$Q_\mu^{(n)} = \begin{pmatrix} \tilde{h}_{0,0} & 0 & \tilde{h}_{0,2} & 0 & \tilde{h}_{0,4} \\ 0 & \tilde{h}_{0,1} & 0 & \tilde{h}_{0,3} & 0 \\ 0 & 0 & \tilde{h}_{1,2} & 0 & \tilde{h}_{1,4} \\ 0 & 0 & 0 & \tilde{h}_{1,3} & 0 \\ 0 & 0 & 0 & 0 & \tilde{h}_{2,4} \end{pmatrix} = \begin{pmatrix} 1 & 0 & \mu & 0 & \mu(\mu-1) \\ 0 & 1 & 0 & \mu & 0 \\ 0 & 0 & 1 & 0 & 2\mu \\ 0 & 0 & 0 & 1 & 0 \\ 0 & 0 & 0 & 0 & 1 \end{pmatrix}.$$

In the following proposition, we will see that the set of GHP is not $D_x - Appel$ sequence (whereas the CHP is $D_x - Appel$ one).

**Proposition 5.** *The GHP fulfills the following relationships*

$$D_x H_{2n}^{(\mu)} = 2n H_{2n-1}^{(\mu)}, \ n \ge 1, \tag{39}$$

$$D_x H_{2n+1}^{(\mu)} = (2n+1) H_{2n}^{(\mu)} + 2n\mu H_{2n-2}^{(\mu+1)}, \ n \ge 1. \tag{40}$$

**Proof.** By deriving (30), we can easily show the result. $\square$

**Remark 1.** *Because of this proposition, we call Generalized Hermite polynomials $D_x-$quasi appel polynomial sequence (for $\mu = 0$, became $D_x-$appel polynomial sequence).*

## 4. The Main Result

*4.1. Eigenvectors and Eigenvalues of the Endomorphism $\Xi^{(h)}$*

By matrix results we can write $M_\mu^{(n)} = Q_\mu^{(n)} \cdot M_H^{(n)} \cdot P_\mu^{(n)} = (c_{i,j})$, $1 \le i,j \le n+1$.
For $i \ge j$ we have

$$c_{i,j} = \begin{cases} -2(i-1) & \text{if } i = j, \\ 0 & \text{if } i > j. \end{cases} \tag{41}$$

For $i < j$ we have

$$c_{i,j} = \begin{cases} \lambda_{\lfloor \frac{i+1}{2} \rfloor, j-1} & \text{if } (j-i) \text{ is even,} \\ 0 & \text{if } (j-i) \text{ is odd,} \end{cases} \tag{42}$$

where $M_H^{(n)}$ is given in (11), and for $1 \le k \le i$

$$\lambda_{k,2i} = \lambda_{k,2i+1} = -4i\mu \frac{(-1)^{i-k}(i-1)!}{(k-1)!}, 1 \le i \le p, p \ge 1. \tag{43}$$

where, for $n = 2p$, one can write

$$M_\mu^{(n)} = \begin{pmatrix} 0 & 0 & \lambda_{1,2} & \cdots & 0 & \lambda_{1,2p} \\ 0 & -2 & 0 & \vdots & \lambda_{1,2p-1} & 0 \\ \vdots & 0 & -4 & \vdots & 0 & \vdots \\ \vdots & \vdots & 0 & \ddots & \vdots & \lambda_{p,2p} \\ \vdots & \vdots & \vdots & 0 & -4p+2 & 0 \\ 0 & 0 & 0 & \vdots & 0 & -4p \end{pmatrix},$$ (44)

**Example 1.** *For $n = 4$:* $M_\mu^{(4)} = Q_\mu^{(4)} \cdot M_H^{(4)} \cdot P_\mu^{(4)}$ *then*

$$M_\mu^{(4)} = \begin{pmatrix} 0 & 0 & \lambda_{1,2} & 0 & \lambda_{1,4} \\ 0 & -2 & 0 & \lambda_{1,3} & 0 \\ 0 & 0 & -4 & 0 & \lambda_{2,4} \\ 0 & 0 & 0 & -6 & 0 \\ 0 & 0 & 0 & 0 & -8 \end{pmatrix} = \begin{pmatrix} 0 & 0 & -4\mu & 0 & 8\mu \\ 0 & -2 & 0 & -4\mu & 0 \\ 0 & 0 & -4 & 0 & -8\mu \\ 0 & 0 & 0 & -6 & 0 \\ 0 & 0 & 0 & 0 & -8 \end{pmatrix},$$

*we get, for example,* $\Xi^{(h)}(H_4^{(\mu)}) = -8H_4^{(\mu)} - 8\mu H_2^{(\mu)} + 8\mu H_0^{(\mu)}$.

**Result 1.** *The Generalized Hermite polynomials fulfill the following relationship*

$$\Xi^{(h)}(H_{2p}^{(\mu)}(x)) = -4pH_{2p}^{(\mu)}(x) - 4p\mu \sum_{k=1}^{p} \frac{(-1)^{p-k}(p-1)!}{(k-1)!} H_{2k-2}^{(\mu)}(x), \ p \geq 1. \quad (45)$$

$$\Xi^{(h)}(H_{2p+1}^{(\mu)}(x)) = -2(2p+1)H_{2p+1}^{(\mu)}(x) - 4p\mu \sum_{k=1}^{p} \frac{(-1)^{p-k}(p-1)!}{(k-1)!} H_{2k-1}^{(\mu)}(x), \ p \geq 1. \quad (46)$$

*which one can write as*

$$\Xi^{(h)}(H_n^{(\mu)}(x)) = -2nH_n^{(\mu)}(x) - 4\mu \left\lfloor \frac{n}{2} \right\rfloor \sum_{k=0}^{\lfloor \frac{n}{2}\rfloor - 1} \frac{(-1)^k(\lfloor \frac{n}{2}\rfloor - 1)!}{(\lfloor \frac{n}{2}\rfloor - 1 - k)!} H_{n-2-2k}^{(\mu)}(x), \ n \geq 2. \quad (47)$$

This result gives the explicit coefficients $\lambda_{k,n}$ satisfying (47).

In this paragraph, we will present the main result in this paper, the spectral vectorial differential equation that the GHP satisfies, which will be given. For that, we recall the endomorphism $\Xi^{(h)} := D_x^2 - 2xD_x$, such as

$$\begin{array}{rcl} \Xi^{(h)} & : & \mathbb{R}_n[X]^{n+1} \to \mathbb{R}_n[X]^{n+1} \\ & & \vec{u}_{k,n} \mapsto \Xi^{(h)}\vec{u}_{k,n} \end{array} \quad (48)$$

where the vector of polynomials $\vec{u}_{k,n} := \begin{pmatrix} P_0(x) \\ P_1(x) \\ \vdots \\ P_k(x) \\ 0 \\ \vdots \\ 0 \end{pmatrix}$ and $\Xi^{(h)}\vec{u}_{k,n} := \begin{pmatrix} \Xi^{(h)}P_0(x) \\ \Xi^{(h)}P_1(x) \\ \vdots \\ \Xi^{(h)}P_k(x) \\ 0 \\ \vdots \\ 0 \end{pmatrix}$, with

$P_k \in \mathbb{R}_n[X]$, for $0 \leq k \leq n$, and $deg(P_k) = k$, then the vector of GHP is given by

$$\vec{h}_{k,n}^{(\mu)} := \begin{pmatrix} H_0^{(\mu)}(x) \\ H_1^{(\mu)}(x) \\ \vdots \\ H_k^{(\mu)}(x) \\ 0 \\ \vdots \\ 0 \end{pmatrix} \in \mathbb{R}_n[X]^{n+1}, 0 \le k \le n. \tag{49}$$

Using (47), one can show the following theorem that we give the matrix of eigenvalue ${}^t M_\mu^{(k,n)}$ of the $\vec{h}_{k,n}^{(\mu)}$, which are the eigenvectors of the endomorphism $\Xi^{(h)}$.

**Theorem 2.** *The vector of GHP $\vec{h}_{k,n}^{(\mu)}$ satisfies the second-order spectral vectorial differential equation as follows*

$$\Xi^{(h)}\vec{h}_{k,n}^{(\mu)} := [D_x^2 - 2xD_x]\vec{h}_{k,n}^{(\mu)} = {}^t M_\mu^{(k,n)}.\vec{h}_{k,n}^{(\mu)}, 0 \le k \le n, \tag{50}$$

*here the $(n+1) \times (n+1)$ matrix $M_\mu^{(k,n)}$ is independent on x is given by*

$$M_\mu^{(k,n)} = \begin{pmatrix} M_\mu^{(k)} & \mathbf{0} \\ ine\mathbf{0} & \mathbf{0} \end{pmatrix}; M_\mu^{(n,n)} = M_\mu^{(n)}, n \ge 0. \tag{51}$$

**Proof.** Using Result 1, the proof of the theorem is immediate.　□

**Remark 2.**

*(1) Using (29), Equation (47) becomes*

$$\Xi^{(h)}(H_{2n}^{(\mu)}(x)) = -4n\left(H_{2n}^{(\mu)}(x) + \mu H_{2n-2}^{(\mu+1)}(x)\right), n \ge 1, \tag{52}$$

$$\Xi^{(h)}(H_{2n+1}^{(\mu)}(x)) = -2(2n+1)H_{2n+1}^{(\mu)}(x) - 4n\mu H_{2n-1}^{(\mu+1)}(x), n \ge 1, \tag{53}$$

*(2) When we take $\mu = 0$ in (47), one can easily see the classical differential Equation (2).*
*(3) Equation (50) is called the second-order spectral vectorial differential equation satisfied by the GHP.*

*4.2. A Pseudo-Spectral Linear Differential Equation of GHP*

In this paragraph, we will transform the spectral vectorial differential equation of GHP (50) into a pseudo-spectral one, where we lose the spectral aspect of this differential equation. In fact, (50) is equivalent to

$$\begin{pmatrix} \Xi^{(h)} H_0^{(\mu)}(x) \\ \Xi^{(h)} H_1^{(\mu)}(x) \\ \vdots \\ \Xi^{(h)} H_k^{(\mu)}(x) \\ 0 \\ \vdots \\ 0 \end{pmatrix} = \begin{pmatrix} {}^t M_\mu^{(k)} & \mathbf{0} \\ ine\mathbf{0} & \mathbf{0} \end{pmatrix} \begin{pmatrix} H_0^{(\mu)}(x) \\ H_1^{(\mu)}(x) \\ \vdots \\ H_k^{(\mu)}(x) \\ 0 \\ \vdots \\ 0 \end{pmatrix}.$$

Therefore, we get $0 \leq j \leq k$

$$\Xi^{(h)} H_j^{(\mu)}(x) = \sum_{i=0}^{k} c_{i+1,j+1} H_i^{(\mu)}(x),$$

where $c_{i,j}$ are given in (41) and (42). This last equation is given explicitly in (47). The even case: Equation (47) is equivalent to (52), then we obtain

$$D^2 H_{2n}^{(\mu)} - 2x D H_{2n}^{(\mu)} = -4n H_{2n}^{(\mu)} - 4n\mu H_{2n-2}^{(\mu+1)}.$$

Using (27) and (39), we get

$$D H_{2n}^{(\mu)} = 2n H_{2n-1}^{(\mu)} = 2nx H_{2n-2}^{(\mu+1)}$$

then

$$D^2 H_{2n}^{(\mu)} - 2x D H_{2n}^{(\mu)} = -4n H_{2n}^{(\mu)} - 2\mu x^{-1} D H_{2n}^{(\mu)}.$$

multiplying by $x^2$, we obtain

$$x^2 D^2 H_{2n}^{(\mu)} - 2x^3 D H_{2n}^{(\mu)} = -4nx^2 H_{2n}^{(\mu)} - 2\mu x D H_{2n}^{(\mu)}.$$

$$x^2 D^2 H_{2n}^{(\mu)} + 2x(-x^2 + \mu) D H_{2n}^{(\mu)} = -4nx^2 H_{2n}^{(\mu)}.$$

**The odd case:** The Equation (47) is equivalent to (53), then we get for $n \geq 1$:

$$D^2 H_{2n+1}^{(\mu)} - 2x D H_{2n+1}^{(\mu)} = -2(2n+1) H_{2n+1}^{(\mu)} - 4n\mu H_{2n-1}^{(\mu+1)}.$$

Using (39), we get $H_{2n-1}^{(\mu+1)} = \frac{1}{2n} D H_{2n}^{(\mu+1)}$, and multiplying by $x^2$ we obtain

$$x^2 D^2 H_{2n+1}^{(\mu)} - 2x^3 D H_{2n+1}^{(\mu)} = -2(2n+1)x^2 H_{2n+1}^{(\mu)} - 2\mu x^2 D H_{2n}^{(\mu+1)}.$$

Using (27) now, $H_{2n}^{(\mu+1)} = \frac{1}{x} H_{2n+1}^{(\mu)}$, we derive this last equality

$$D H_{2n}^{(\mu+1)} = \frac{1}{x} D H_{2n+1}^{(\mu)} - \frac{1}{x^2} H_{2n+1}^{(\mu)} >$$

After some simplification, we get

$$x^2 D^2 H_{2n+1}^{(\mu)} + 2x(-x^2 + \mu) D H_{2n+1}^{(\mu)} = -2((2n+1)x^2 - \mu) H_{2n+1}^{(\mu)}.$$

We can conclude, for $n \geq 0$, that the GHP $\{H_n^{(\mu)}\}_{n \geq 0}$ verifies the following pseudo-spectral linear differential equation [5] (Problem 25 page 380):

$$x^2 D_x^2 H_n^{(\mu)} + 2x(-x^2 + \mu) D_x H_n^{(\mu)} + 2(nx^2 + \mu\epsilon) H_n^{(\mu)} = 0; \quad \epsilon = \begin{cases} 0, & \text{if } n \text{ is even;} \\ -1, & \text{if } n \text{ is odd.} \end{cases} \tag{54}$$

## 5. Conclusions

In this work, we were successful at giving a second-order spectral vectorial differential equation (SVDE) that GHP satisfies, where one can apply it to different areas. In addition, we use this SVDE to give a simple proof of the pseudo-spectral differential equation satisfied by GHP given in [5] (Problem 25 p. 380).

Conjecture

One case of a nonsymmetric semi-classical polynomials of class $s = 1$: An example of Generalized Jacobi polynomials $\{J_n^{(\alpha,\alpha+1)}(x,\mu)\}_{n \geq 0}$ that we denote by $J_n^{(\alpha,\alpha+1)}(x,\mu) :=$

$J_n^{(\alpha,\mu)}(x)$ is given in [20]. This sequence is orthogonal with respect to the positive weight $w(x,\alpha;\mu) =\mid x \mid^{-\mu} (1-x^2)^\alpha(1-x), \mu < 1, \alpha > -1$ on $[-1,1]$. This generalized Jacobi polynomials sequence fulfils the following TTRR

$$J_0^{(\alpha,\mu)}(x) = 1, \qquad J_1^{(\alpha,\mu)}(x) = x - \beta_0,$$
$$J_{n+2}^{(\alpha,\mu)}(x) = (x - \beta_{n+1})J_{n+1}^{(\alpha,\mu)}(x) - \gamma_{n+1}J_n^{(\alpha,\mu)}(x), \tag{55}$$

with the coefficients $\beta_n$ and $\gamma_n$ are given by [20]

$$\beta_0 = -\frac{\mu - 1}{\mu - 2\alpha - 3}, \tag{56}$$

$$\beta_n = (-1)^n \frac{\mu(\mu - 2n - 2\alpha - 4) + (-1)^{n+1}(2\alpha + 1)}{(2n + 2\alpha + 3 - \mu)(2n + 2\alpha + 5 - \mu)}, \tag{57}$$

$$\gamma_{2n+1} = 2\frac{(n + \alpha + 1)(2n + 1 - \mu)}{(4n + 2\alpha + 3 - \mu)^2}, \tag{58}$$

$$\gamma_{2n+2} = \frac{(2n + 2)(2n + 2\alpha + 3 - \mu)}{(4n + 2\alpha + 5 - \mu)^2}. \tag{59}$$

For $\beta = \alpha + 1$ in the $\Xi^{(j)}$ defined in (5), we get

$$\Xi^{(j)} := (x^2 - 1)D_x^2 + ((2\alpha + 3)x + 1)D_x; \lambda_n^{(j)} = n(n + 2\alpha + 2).$$

We set $\vec{j}_{k,n}^{(\alpha,\mu)} := \begin{pmatrix} J_0^{(\alpha,\mu)}(x) \\ J_1^{(\alpha,\mu)}(x) \\ \vdots \\ J_k^{(\alpha,\mu)}(x) \\ 0 \\ \vdots \\ 0 \end{pmatrix}$ and $\Xi^{(j)}\vec{j}_{k,n}^{(\alpha,\mu)} = \begin{pmatrix} \Xi^{(j)}J_0^{(\alpha,\mu)}(x) \\ \Xi^{(j)}J_1^{(\alpha,\mu)}(x) \\ \vdots \\ \Xi^{(j)}J_k^{(\alpha,\mu)}(x) \\ 0 \\ \vdots \\ 0 \end{pmatrix}.$

In the following theorem, we give the $(n + 1) \times (n + 1)$ matrix of eigenvalue ${}^tJ_{\alpha,\mu}^{(k,n)}$ of the $\vec{j}_{k,n}^{(\alpha,\mu)}$, which are the eigenvectors of the endomorphism $\Xi^{(j)}$.

**Conjecture**

The vector $\vec{j}_{k,n}^{(\alpha,\mu)}$ satisfies the second-order spectral vectorial differential equation as follows

$$\Xi^{(j)}\vec{j}_{k,n}^{(\alpha,\mu)} := [(x^2 - 1)D_x^2 + ((2\alpha + 3)x + 1)D_x]\vec{j}_{k,n}^{(\alpha,\mu)} = {}^tJ_{\alpha,\mu}^{(k,n)} \cdot \vec{j}_{k,n}^{(\alpha,\mu)}, \tag{60}$$

then $\overrightarrow{j_{\alpha,\mu}^{(n)}}$ are eigenvectors of the endomorphism $\Xi^{(j)}$, where the matrix ${}^tJ_{\alpha,\mu}^{(n)}$ of eigenvalue given by the $(n + 1) \times (n + 1)$ matrix $J_{\alpha,\mu}^{(k,n)}$

$$J_{\alpha,\mu}^{(k,n)} = \begin{pmatrix} J_{\alpha,\mu}^{(k)} & \mathbf{0} \\ ine\mathbf{0} & \mathbf{0} \end{pmatrix}; J_{\alpha,\mu}^{(n,n)} = J_{\alpha,\mu}^{(n)}, n \geq 0, \tag{61}$$

where $J_{\alpha,\mu}^{(k)} = (c_{i,l})$, $1 \leq i, l \leq k + 1$ and $0 \leq k \leq n$ with

For $i \geq l$: $c_{i,l} = \begin{cases} (i - 1)(i + 2\alpha + 1) & \text{if } i = l. \\ 0 & \text{if } i > l. \end{cases}$

For $i < l$ : $c_{i,l} = \lambda_{l-i}^{(l-1)}$ (see below).

As a consequence of this conjecture, one can deduce the following differential equations:

**The even case ($n = 2p$):**

$$\Xi^{(j)} J_{2p}^{(\alpha,\mu)}(x) = \lambda_{2p}^{(j)} J_{2p}^{(\alpha,\mu)}(x) + \mu\big(\sum_{k=1}^{p} \lambda_{2k}^{(2p)} J_{2p-2k}^{(\alpha,\mu)}(x) + \sum_{k=0}^{p-1} \lambda_{2k+1}^{(2p)} J_{2p-2k-1}^{(\alpha,\mu)}(x)\big)$$

with

$$\lambda_{2k}^{(2p)} = \frac{(-1)^k k!\binom{p}{k}(2\alpha+2-\mu+4p-2k)}{(p+\alpha+1)^{\overline{-k}}(\alpha+2p+\frac{3}{2}-\frac{\mu}{2}-2k)^{\overline{2k}}}$$

$$\lambda_{2k+1}^{(2p)} = \frac{(-1)^{k+1}(2k+1)(k+1)!\binom{p}{k+1}}{(p+\alpha+1)^{\overline{-k}}(\alpha+2p+\frac{1}{2}-\frac{\mu}{2}-2k)^{\overline{2k+1}}}$$

**The odd case ($n = 2p + 1$):**

$$\Xi^{(j)} J_{2p+1}^{(\alpha,\mu)}(x) = \lambda_{2p+1}^{(j)} J_{2p+1}^{(\alpha,\mu)}(x) + \mu\big(\sum_{k=0}^{p} \lambda_{2k+1}^{(2p+1)} J_{2p-2k}^{(\alpha,\mu)}(x) + \sum_{k=1}^{p} \lambda_{2k}^{(2p+1)} J_{2p-2k+1}^{(\alpha,\mu)}(x)\big)$$

with

$$\lambda_{2k+1}^{(2p+1)} = \frac{(-1)^k(2k+1)k!\binom{p}{k}}{(p+\alpha+2)^{\overline{-k-1}}(\alpha+2p+\frac{3}{2}-\frac{\mu}{2}-2k)^{\overline{2k+1}}}$$

$$\lambda_{2k}^{(2p+1)} = \frac{(-1)^k k!\binom{p}{k}(2\alpha+4-\mu+4p-2k)}{(p+2+\alpha)^{\overline{-k}}(\alpha+2p+\frac{5}{2}-\frac{\mu}{2}-2k)^{\overline{2k}}}.$$

**Author Contributions:** Writing—review & editing, M.J.A. and M.B. All authors have read and agreed to the published version of the manuscript.

**Funding:** This research was partially funded by Qassim Univesity.

**Institutional Review Board Statement:** Not applicable.

**Informed Consent Statement:** Not applicable.

**Data Availability Statement:** Not applicable.

**Acknowledgments:** Special thanks to Milovanovich G., who has guided the author to submit the proper topic of the journal or to the reviewer who has considerably improved the presentation of this paper.

**Conflicts of Interest:** The author declares no conflict of interest.

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
