# Peer review of "On Spectral Vectorial Differential Equation of Generalized Hermite Polynomials"

_axioms, doi:10.3390/axioms11070344_

Round 1

Reviewer 1 Report

Dear Editor

In this article, the authors presented the normalized generalized symmetric Hermite polynomials (GHP) orthogonal with respect to the positive and symmetric weight and proved some results which formulated as the second-order spectral vectorial differential equation SVDE that the GHP satisfy. This SVDE is different to the one given in G. Szego (problem 25. p380) which is a pseudo-spectral equation. Secondly, they provided the SVDE, as conjecture, that a nonsymmetric Generalized Jacobi polynomials satisfy.

Main results seems correct but have several gaps, introduction needs improvements, language is poor, it need improvement. Several grammatical and language mistakes, like "if it exits page 1, real {\lemda_{n}}, cases instead of case, page 2: property instead of propriety, and al., and several more...

Missing terms \nu?, normalized generalized, page 3: This polynomials..... a lot more... alike [?], type[7] etc.....

proof of proposition 3, 5must be explained in details:

Result6 and Example 7 must be added in proof?

Example 10 must be explained in details Conclusion and conjecture must be improved.

Cite the following papers in introduction:

Convergence of Generalized Quasi-Nonexpansive Mappings in Hyperbolic Space

Interpolative C´iric´-Reich-Rus-type best proximity point results with applications

Single and Multivalued Maps on Parametric Metric Spaces Endowed with an Equivalence Relation

Author Response

Introduction needs improvements,  Done

language is poor, it need improvement. Done

Several grammatical and language mistakes, like "if it exits page 1, real {\lemda_{n}}, cases instead of case, page 2: property instead of propriety, and al., and several more... Done

Missing terms \nu?, normalized generalized, page 3: This polynomials..... a lot more... alike [?], type[7] etc..... Done

proof of proposition 3, 5must be explained in details: done

Result6 and Example 7 must be added in proof? Done

Cite the following papers in introduction:

For these papers, I apologise to the referee. These papes are very difficult to me and I can not know where I can cite them in my paper:

<<Convergence of Generalized Quasi-Nonexpansive Mappings in Hyperbolic Space

Interpolative C´iric´-Reich-Rus-type best proximity point results with applications

Single and Multivalued Maps on Parametric Metric Spaces Endowed with an Equivalence Relation>>

Reviewer 2 Report

In the present study, the authors focused their attention on the normalized generalized symmetric Hermite polynomials. The presentation of the study is generally appropriate and the procedures seem correct. Out of curiosity after reading the manuscript, I scanned it through the iThenticate system and have seen that there are some text and equation overlaps between this manuscript and other papers published in literature. For the information presented in some parts of the submission, the authors had to have cited the relevant source. I think the authors should check out the two studies that I have listed below.

* S. Belmehdi, "Generalized Gegenbauer orthogonal polynomials", Journal of Computational and Applied Mathematics 133 (2001) 195–205

* P. Maroni and T. A. Mesquita, "Appell polynomial sequences with respect to some differential operators", e-Print: 1404.3615 [math.CA].

To summarize, unfortunately, it is not possible for me to recommend the publication of the work in this form. This study requires a major revision.

Author Response

The authors had to have cited the relevant source. I think the authors should check out the two studies that I have listed below.

S. Belmehdi, "Generalized Gegenbauer orthogonal polynomials", Journal of Computational and Applied Mathematics 133 (2001) 195–205

* P. Maroni and T. A. Mesquita, "Appell polynomial sequences with respect to some differential operators", e-Print: 1404.3615 [math.CA].  Done

To summarize, unfortunately, it is not possible for me to recommend the publication of the work in this form. This study requires a major revision.

Done

Reviewer 3 Report

The content of this paper is fine, original, and interesting

However the presentation is rather poor and need to be improved before the paper can be accepted.

The English usage and grammar needs serious revision.

In particular:

1) The word 'aspect' in the title is not acceptable. It may be that instead of 'The spectral aspectral' (which means The spectral look) the authors means 'Spectral aspepects...' It seems a small change but it makes a lot of diffenetence in the common usage of english!

2) Same with word 'deal' in the abstract

3) Th abstract is very bad, and non-informative. It should be fostered and improved, including a short review of resuts and conclusions

4) change 'case' to plural 'cases' before Eq. (2)

5) 'Laguerre' after eq (2) is misspelled

6)  Are the adresses complete?

In page 2

7) Put COP and SCOP between parenthesis

8) property is misspelled in line 7

9) In line 8 it is stated '...des not remain valid' Explain why (that happens in other places along the text

10) In line 13 substitute 'and' by 'et'

11) In line 15 the authors write 'The authors found' It is not clear if they refere to themselves in this paper or Marcellan et al. Please, clarify

12) lines 22-23 I found the reference written in a wird, non-standard way. Pease, chane

13) The grammar of paragraph in lines 21-26 should be improved. I understand what they mean, but this is not standard english

14) In line 29 change 'We need to remember' by 'Recall'

Page 3

15) Is 'monic' a well defined term?

16) After eq. (11) change (GHP) to GHP

Page 4

17) what does 'semi-classical' means? Please explain

18) Reference in line 19 is missing !!

Page 5 

19) Lines 1 and 2 properties is misspelled and Polynomial should be in lowercase

Page 7

20)  Acronisms 'TTRR' is not defined

Page 9

21) In 'Result 11' the acronism GHP is not used

Author Response

The presentation is  improved.

The English usage is revised.

In particular:

1) The word 'aspect' in the title is not acceptable. The title becomes

"On spectral vectorial differential equation of generalized Hermite polynomials"

2) Same with word 'deal' in the abstract "In this paper we give, first, some results on monic generalized
Hermite polynomials (GHP) $\{H_n^{(\mu)}(x)\}_{n\geq 0}$ orthogonal
with respect to the positive weight $| x |^{2\mu}e^{-x^2}, \
\mu>-\frac{1}{2},\ x\in\mathbb{R}$ which will lead to formulate the
second-order {\bf spectral} vectorial differential equation SVDE
that the GHP satisfy. This SVDE differs from the one given in G.
Szego (problem 25. p380) which is a {\bf pseudo-spectral} equation.
Second, we give the SVDE, as conjecture, satisfied by the
generalized Jacobi polynomials $J_n^{(\alpha,\alpha+1)}(x,\mu)$,
orthogonal with respect the positive weight $w(x,\alpha;\mu)=|
x|^{-\mu}(1-x^2)^\alpha(1-x),\ \mu<1,\ \alpha>-1$ on $[-1,1]$."

3) Th abstract is very bad, and non-informative. It should be fostered and improved, including a short review of resuts and conclusions

4) change 'case' to plural 'cases' before Eq. (2) (done)

5) 'Laguerre' after eq (2) is misspelled (done)

6)  Are the adresses complete? (done)

In page 2

7) Put COP and SCOP between parenthesis (done)

8) property is misspelled in line 7 (done)

9) In line 8 it is stated '...des not remain valid' Explain why (that happens in other places along the text  (done)

10) In line 13 substitute 'and' by 'et' (done)

11) In line 15 the authors write 'The authors found' It is not clear if they refere to themselves in this paper or Marcellan et al. Please, clarify (done)

12) lines 22-23 I found the reference written in a wird, non-standard way. Pease, chane (done)

13) The grammar of paragraph in lines 21-26 should be improved. I understand what they mean, but this is not standard english (done)

14) In line 29 change 'We need to remember' by 'Recall' (done)

Page 3

15) Is 'monic' a well defined term? (done)

16) After eq. (11) change (GHP) to GHP (done)

Page 4

17) what does 'semi-classical' means? Please explain (done)

18) Reference in line 19 is missing !! (done)

Page 5 

19) Lines 1 and 2 properties is misspelled and Polynomial should be in lowercase (done)

Page 7 

20)  Acronisms 'TTRR' is not defined (done)

Page 9

21) In 'Result 11' the acronism GHP is not used (done)

Round 2

Reviewer 1 Report

Agreed with the changes. In current article seems good, and it can be accepted but the authors should properly cite all of the listed articles in the previous report.

Reviewer 2 Report

It has been seen that the author has made the relevant resource updates. However, I have two more suggestions, which can be corrected with a minor revision. 

* In page 4, the author should cite Ref[1] at the end of the sentence written before equation (2.1).

* In page 4, at the end of the sentence written before equation (2.6), the reference number does not seem. There is a question mark in the form of [?] and this part in the tex file needs to be corrected. It seems that the relevant resource there is Ref[1].

The revised article with these corrections does not need to be sent back to me. After this minor revision, I think the work could be accepted in the journal.

Reviewer 3 Report

The authors have adequately taken into account my comments and criticism, and the paper has improved.

I can recommend its publicaton 

This manuscript is a resubmission of an earlier submission. The following is a list of the peer review reports and author responses from that submission.